# PARIS, an optogenetic method for functionally mapping gap junctions

Ling Wu[1,2,3], Ao Dong[1,2,3], Liting Dong[3], Shi-Qiang Wang[1], Yulong Li[1,2,3,4]*

[1]State Key Laboratory of Membrane Biology, Peking University School of Life Sciences, Beijing, China; [2]PKU-IDG/McGovern Institute for Brain Research, Beijing, China; [3]Peking-Tsinghua Center for Life Sciences, Beijing, China; [4]Chinese Institute for Brain Research, Beijing, China

**Abstract** Cell-cell communication via gap junctions regulates a wide range of physiological processes by enabling the direct intercellular electrical and chemical coupling. However, the in vivo distribution and function of gap junctions remain poorly understood, partly due to the lack of non-invasive tools with both cell-type specificity and high spatiotemporal resolution. Here, we developed PARIS (pairing actuators and receivers to optically isolate gap junctions), a new fully genetically encoded tool for measuring the cell-specific gap junctional coupling (GJC). PARIS successfully enabled monitoring of GJC in several cultured cell lines under physiologically relevant conditions and in distinct genetically defined neurons in *Drosophila* brain, with ~10 s temporal resolution and sub-cellular spatial resolution. These results demonstrate that PARIS is a robust, highly sensitive tool for mapping functional gap junctions and study their regulation in both health and disease.

DOI: https://doi.org/10.7554/eLife.43366.001

## Introduction

Gap junctions are intercellular channels that are expressed in virtually all tissue types in both vertebrates and invertebrates. They allow direct transfer of small molecules such as ions, metabolites and second messengers between connected cells, thereby playing essential roles in various physiological processes, including embryo development, metabolic coordination of avascular organs, tissue homeostasis and synchronizing electrical activity in excitable tissues (*Bennett and Zukin, 2004*; *Kumar and Gilula, 1996*). Defects of gap junction function are linked with a wide array of diseases, including myocardial infarction, hearing loss and hypomyelinating disorder (*Jongsma and Wilders, 2000*; *Laird, 2010*; *Söhl et al., 2005*). Studying gap junction coupling (GJC) under physiological or disease conditions in complex systems such as the nervous system requires a non-invasive method with both cell-type specificity and high spatiotemporal resolution. Current methods used to monitor GJC include paired electrophysiological recordings (*Bennett et al., 1963*; *Furshpan and Potter, 1959*), dye microinjection (*Stewart, 1978*), fluorescence recovery after photobleaching (FRAP) (*Wade et al., 1986*), and local activation of molecular fluorescent probes (LAMP) (*Dakin et al., 2005*), all of which are either invasive and/or lack cell-type specificity, thus limiting their use in heterogeneous tissues. Recently, hybrid approaches have been developed in which genetically encoded hydrolytic enzymes or promiscuous transporters are used to introduce small molecule substrates or peptides tagged with a fluorescent label in specific cells (*Qiao and Sanes, 2015*; *Tian et al., 2012*). Although these methods might provide cell-type-specific investigation of GJC, the requirement of exogenous substrate and the background generated by endogenous enzymes or transporters still make them difficult to apply in vivo. In addition, dye diffusion is an irreversible process and thus these methods are difficult to be applied to examine the same gap junctions repeatedly in order to measure their dynamics, regulations and plasticity (*Dong et al., 2018*).

*For correspondence:
yulongli@pku.edu.cn

Competing interests: The authors declare that no competing interests exist.

**eLife digest** For the tissues and organs of our bodies to work properly, the cells within them need to communicate with each other. One important part of cellular communication is the movement of signals – usually small molecules or ions – directly from one cell to another. This happens via structures called gap junctions, a type of sealed 'channel' that connects two cells.

Gap junctions are found throughout the body, but investigating their precise roles in health and disease has been difficult. This is due to problems with the tools available to detect and monitor gap junctions. Some are simply harmful to cells, while others cannot be restricted to specific cell populations within a tissue. This lack of specificity makes it difficult to study gap junctions in the brain, where it is important to understand the connectivity patterns between distinct types of nerve cells. Wu et al. wanted to develop a new, non-harmful method to track gap junctions in distinct groups of cells within living tissues.

To do this, Wu et al. devised PARIS, a two-part, genetically encoded system. The first part comprises a light-sensitive molecular 'pump', which can only be turned on by shining a laser onto the cell of interest. When the pump is active, it transports hydrogen ions out of the cell. The second part of the system is a fluorescent sensor, present inside 'receiving' cells, which responds to the outcoming hydrogen ions (small enough to pass through gap junctions). If an illuminated 'signaling' cell is connected via gap junctions to cells containing the fluorescent sensor, they will light up within seconds, but other cells not connected through gap junctions will not.

The researchers first tested PARIS in cultured human and rat cells that had been genetically engineered to produce both components of the system. The experiments confirmed that PARIS could both detect networks of gap junctions in healthy cells and reveal when these networks had been disrupted, for instance by drugs or genetic mutations. Experiments using fruit flies demonstrated that PARIS was stable in living tissue and could also map the gap junctions connecting specific groups of nerve cells.

PARIS is a valuable addition to the toolbox available to study cell communication. In the future, it could help increase our understanding of diseases characterized by defective gap junctions, such as seizures, cardiac irregularities, and even some cancers.

DOI: https://doi.org/10.7554/eLife.43366.002

To overcome these limitations, we developed an all-optical approach named PARIS (<u>p</u>airing <u>a</u>ctuators and <u>r</u>eceivers to optically <u>is</u>olate gap junctions) in which we express an optically controlled actuator in one cell, to generate an electrochemical gradient of specific molecules between two connected cells, and a fluorescent receiver in the adjacent cell, to detect the movement of the molecules across the gap junctions. GJC between the actuator cell (i.e. expressing actuators) and the receiver cell (i.e. expressing receivers) is detected by a fluorescence increase in the receiver following the optical activation of the actuator (*Figure 1A*).

## Results

### Development of a novel all-optical tool for monitoring GJC

At the beginning, we tested several pairs of optical actuators/receivers based on generating/detecting small molecules that can readily diffuse across gap junctions, such as cGMP, $Ca^{2+}$ and proton ($H^+$). Our first step was to test whether the actuator/receiver pair can generate a cell-autonomous signal. We found that, when co-expressed in HEK293T cells (i.e. in the *cis* configuration), neither a light-activated cGMP cyclase BeCylOp (*Gao et al., 2015*) paired with a cGMP sensor FlincG3 (*Bhargava et al., 2013*) nor the red shifted channelrhodopsin CsChrimson (*Klapoetke et al., 2014*) paired with a sensitive $Ca^{2+}$ indicator GCaMP6s (*Chen et al., 2013*) could generate detectable light-induced signal (*Figure 1—figure supplement 1*).

Interestingly, when we co-expressed a light-gated outward proton pump ArchT (*Han et al., 2011*) and a pH-sensitive green fluorescent protein pHluorin (*Miesenböck et al., 1998*; *Sankaranarayanan et al., 2000*) in HEK293T cells, a 4 s laser illumination at 561 nm elicited a robust increase in pHluorin fluorescence, with the membrane-targeted pHluorin (pHluorinCAAX) producing

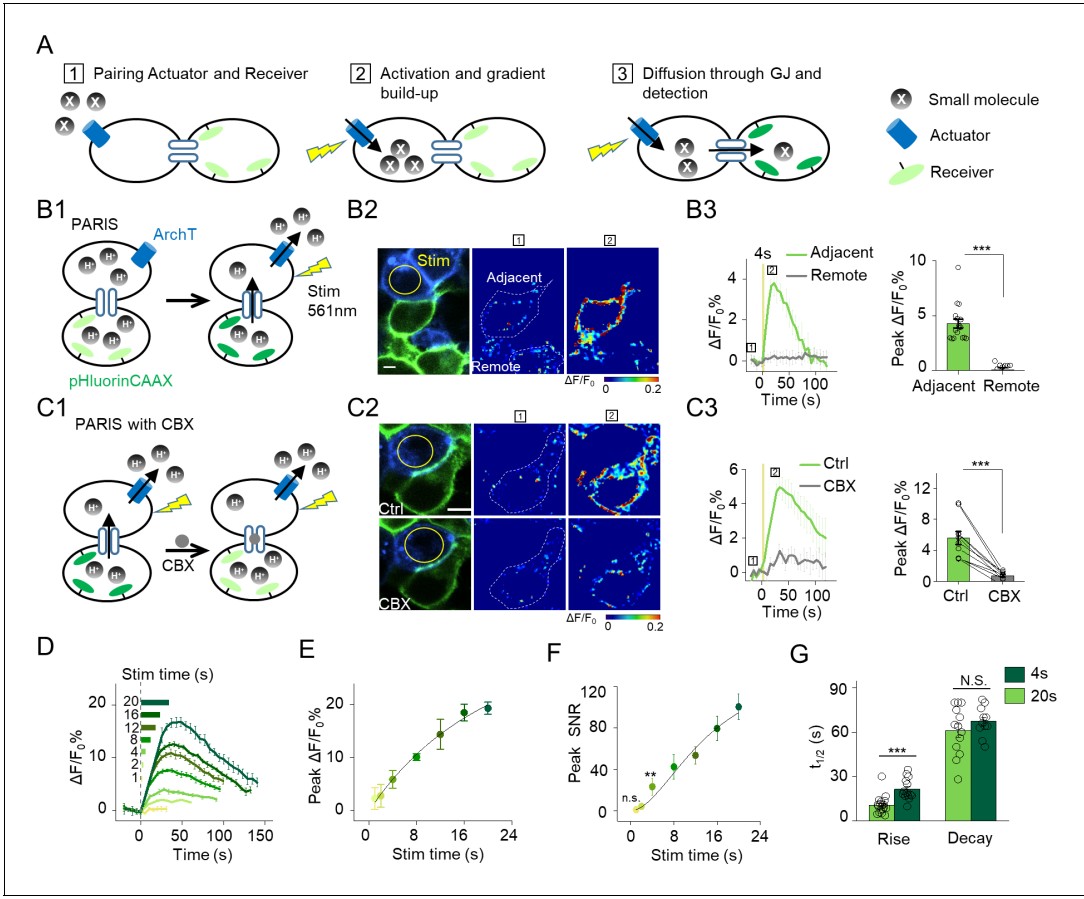

**Figure 1.** The principle behind PARIS and proof-of-concept in cultured cells by PARIS based on ArchT and pHluorin. (**A**) Schematic diagram depicting the principle and process of PARIS. (**B1–B3**) Demonstration of PARIS using in HEK293T cells. (**B1**) Schematic depicting the principle of ArchT/pHluorin pair based PARIS. (**B2**) Representative images showing expression of the actuator ArchT-BFP (blue) and the receiver pHluorinCAAX (green) in transfected HEK293T cells neighboring to each other. The pseudocolor images showing the basal fluorescence and 4 s ArchT activation induced pHluorin response in the adjacent cell/remote cell. The boxed '1' and '2' above the images (or traces in B3) identify stages before photostimulation and in the peak response. (**B3**) Representative traces and group analysis of $\Delta F/F_0$ in the cells adjacent and remote to the actuator cells (n = 10–17 cells). The stimulus (561 nm light, 0.5 mW) is indicated by the yellow circle in the image and the yellow vertical line in the traces. (**C1–C3**) Similar to (**B**), except PARIS signals were recorded before and after CBX treatment (100 µM, 10 min, n = 10 cells). (**D**) Representative traces of $\Delta F/F_0$ using increasing stimulation time (laser power, 0.5 mW). (**E–F**) Grouped peak $\Delta F/F_0$ and peak signal-to-noise ratio (SNR) of PARIS signals recorded with increasing stimulation time (n = 5–12 cells for each data point); the data were fit to a single Hill function (solid lines). (**G**) Summary of the half-rise and half-decay times of the PARIS signals measured with 4 s or 20 s stimulation (n = 14–18 cells). The scale bars represent 10 µm. *p<0.05, **p<0.01, ***p<0.001, N.S., not significant (p>0.05). In this and subsequent figures, error bars in the representative traces indicate the s.e.m. from three repeat experiments.

DOI: https://doi.org/10.7554/eLife.43366.003

The following figure supplements are available for figure 1:

**Figure supplement 1.** Poor performance of actuators and receivers based on detecting cGMP and $Ca^{2+}$.
DOI: https://doi.org/10.7554/eLife.43366.004

**Figure supplement 2.** Functional characterization of actuator and receiver based on ArchT and pHluorin.
DOI: https://doi.org/10.7554/eLife.43366.005

**Figure supplement 3.** Probing GJC propagation within connected HEK293T cells by PARIS.
DOI: https://doi.org/10.7554/eLife.43366.006

**Figure supplement 4.** Quantification of ArchT activation-induced pH change in actuator cells.
DOI: https://doi.org/10.7554/eLife.43366.007

a larger change in fluorescence than the cytosolic pHluorin (*Figure 1—figure supplement 2A,B*). No light-induced change in fluorescence was observed in cells that co-expressing pHluorinCAAX and the deficient proton-pump ArchTD95N (*Kralj et al., 2011*), or in cells that only express pHluorinCAAX (*Figure 1—figure supplement 2A,B*). Furthermore, the evoked response is dependent on

both the duration and the power of the activating light (*Figure 1—figure supplement 2C–F*). These results demonstrate that ArchT and pHluorin can function as a pair of proton actuator and proton sensor.

We next examined whether PARIS based on ArchT/pHluorin can be used to measure GJC between cultured HEK293T cells, which endogenously express both connexin (Cx) 43 and Cx45, therefore spontaneously form gap junctions between adjacent cells (*Butterweck et al., 1994*; *Langlois et al., 2008*). When ArchT and pHluorin were separately expressed in neighboring cells (i.e. in the *trans* configuration, see Materials and methods; *Figure 1B1*), a brief photoactivation of ArchT in the actuator cells (4 s,~0.5 mW, indicated by the yellow circle in *Figure 1B2*) faithfully induced a ~ 4.3% $\Delta F/F_0$ increase in pHluorinCAAX fluorescence in the neighboring receiver cells whereas non-adjacent pHluorinCAAX-expressing cells had no measurable change in fluorescence (*Figures 1B2–B3*). Application of carbenoxolone (CBX, 100 µM) which blocks gap junctions (*Connors, 2012*) significantly decreased the light-induced PARIS signal (*Figure 1C*), confirming that the signal measured in receiver cells is mediated by GJC. Similar to autonomous signals, increasing the duration of the illumination pulse from 1 s to 20 s incrementally increased the PARIS response from ~2% to~20% (*Figure 1D–E*). A 4 s laser pulse was sufficient to induce a robust PARIS signal (SNR = 23 ± 8, *Figure 1F*) with a half-rise time of ~10 s (*Figure 1G*). On the other hand, a 20 s laser pulse induced an ~4.3-fold increase in the signal-to-noise ratio compared to 4 s with a half-rise time of ~21 s (*Figure 1F,G*); however, the half-decay time did not differ between a 4 s pulse and a 20 s pulse ($t_{1/2\ decay}$ = 61 ± 5s and 67 ± 3 s respectively, *Figure 1G*). We also observed the spatially graded PARIS signals in three receiver cells that are sequentially connected to the actuator cell (*Figure 1—figure supplement 3*). Specifically, the directly connected cell had the strongest response, and the thirdly connected cell had the weakest response (*Figure 1—figure supplement 3D*).

We then quantified the ArchT-induced pH change in the actuator cells using the ratiometric pH indicator mTagBFP-pHluorinCAAX generated by fusing the pH-insensitive blue fluorescent protein mTagBFP (*Subach et al., 2008*) to the N-terminus of pHluorinCAAX and then calibrating the correlation between pH and the ratio of GFP/BFP fluorescence (*Figure 1—figure supplement 4*). Based on a fit to the titration curve, we estimated that a 4 s and 20 s laser pulse induces a transient increase of intracellular pH from 7.35 to 7.45 and 7.80 respectively in actuator cells (*Figure 1—figure supplement 4D–F*), which allowed us to repeatedly elicit a PARIS signal in specific cells as shown above. Together, these data provide proof-of-principle that PARIS is a robust tool for measuring GJC between connected cells.

## Electrophysiological validation of PARIS and its comparison with FRAP in HEK293T cells

We have showed that PARIS could detect GJC in a photostimulation-dependent way and sensitive to CBX (*Figure 2A,D1* and *Figure 1*). Next, we further validated PARIS by patch-clamping the receiver cell in order to record the gap junction-mediated current induced by activating the actuator cell using a laser pulse (*Figure 2B1*). Applying increasingly stronger light pulses to the actuator cell yielded time-locked currents in the receiver cell that were blocked by CBX (*Figure 2B2,D2*). In the same cells, voltage steps on the actuator cell also elicited non-rectifying and CBX sensitive currents in the receiver cell (*Figure 2C,D3*). Quantification of the group data showed that the CBX inhibition of GJC was independent from the approaches used to activate the actuator cell (by light or voltage) and from the signals measured in the receiver cell (pHluorinCAAX fluorescence or currents) (*Figure 2E*). In addition, we performed a head-to-head comparison between PARIS and FRAP—a dye diffusion based methods which detects the gap junction mediated fluorescence recovery after photobleaching (*Figure 2F*). The PARIS signal was stable for five sequential pulses at 2 min intervals, whereas the FRAP signal decayed considerably over the same time period in terms of both basal fluorescence and SNR (*Figure 2G–I*). Moreover, the PARIS signal had considerably faster kinetics than FRAP, with a half-rise time of ~21 s compared to ~197 s, respectively (*Figure 2J*).

## PARIS enables reporting regulations of gap junction and disease-causing mutations in connexin genes

Phosphorylation has been implicated in the modulation of GJC by affecting the trafficking, assembly/disassembly, degradation and gating of gap junctions (*Laird, 2005*; *Nihei et al., 2010*). To test

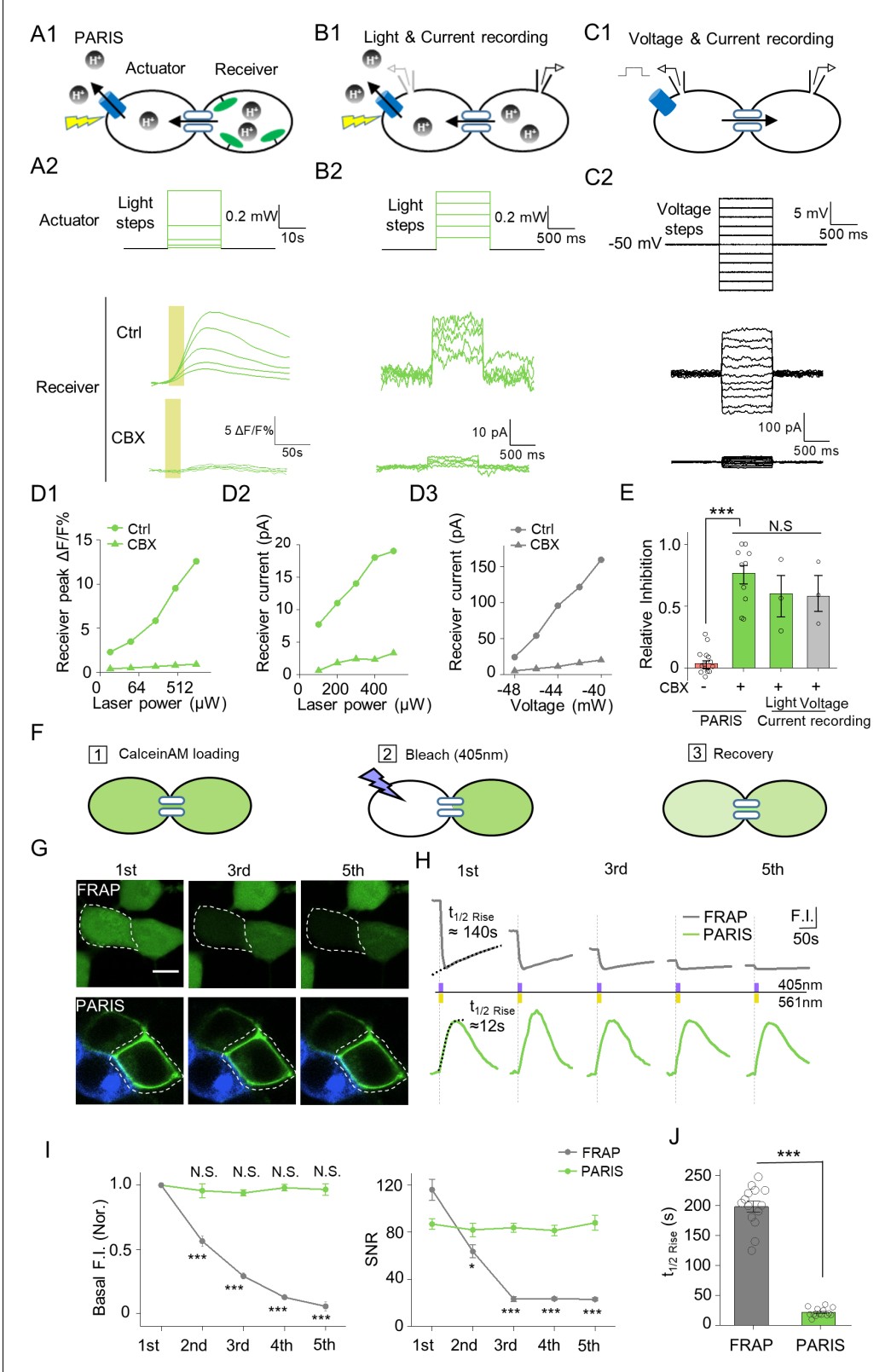

**Figure 2.** PARIS's validation by electrophysiological recording and its comparison with FRAP in HEK293T cells. (A1–A2) PARIS detection of gap junctional coupling under increasing light power and the application of CBX (0.01 mW to 1.5 Mw, 20 s). (B1–B2) Electrophysiological recording of the gap junctional currents during actuator activation. (B1), Schematic diagram depicting dual patch-clamp recording of a pair of HEK293T cells connected by

*Figure 2 continued*

gap junctions; one cell of the pair expresses ArchT-BFP. (**B2**), Light steps applied to the actuator cell (top, from 0.1 mW to 0.5 mW), recorded currents in the receiver cell (middle); elimination of the currents by the application of CBX (100 µM, 10 min treatment, bottom). (**C1–C2**) Similar experiment using the same cells shown in (**B**), except that voltage steps (from −60 mV to −40 mV) were applied to the actuator cell, while the receiver cell was clamped at −50 mV. (**D1–D3**) Input-output curve for peak $\Delta F/F_0$ % or currents measured in the receiver cell versus illumination intensity or voltage in the absence or presence of CBX. (**E**) Summary data showing the relative CBX-mediated inhibition of signals measured with PARIS or current recording (n = 3–15 cells per group). (**F**) Schematic depicting the process of FRAP method to detect gap junctional communication using Calcein-AM dye. (**G,H**) Comparison of basal fluorescence between PARIS and FRAP during sequential five photostimulation and photobleaching. (**G**) Exemplary images showing the fluorescence of Calcein or pHluorin at the beginning of 1st, 3rd and 5th FRAP or PARIS measurement. (**H**) Traces of fluorescence intensity from five consecutive FRAP and PARIS measurements. The half-rise time for the first traces are indicated with dotted lines reflecting the curve-fit analysis. Yellow or purple lines represent the stimulation (561 nm) or bleaching (405 nm) period. (**I**) Quantified comparison of basal fluorescence, SNR and the half-rise time between FRAP and PARIS method as experiments shown in (**F,G**) (n = 5 cells). The scale bar represents 10 µm in (**G**). ***p<0.001, N.S., not significant.

DOI: https://doi.org/10.7554/eLife.43366.008

whether PARIS could report GJC under different regulations such as protein phosphorylation, we treated PARIS-expressing HEK293T cells with the cAMP analog 8-Br-cAMP, the adenylyl cyclase agonist forskolin, or the protein kinase C (PKC) agonist tetradecanoylphorbol acetate (TPA). Compared with the control group, treating cells with TPA significantly inhibited the PARIS signal compared to the control group; in contrast, neither 8-Br-cAMP nor forskolin had obvious effect (*Figure 3A,B*), suggesting that activating PKC—but not protein kinase A—inhibits GJC, which is in general consistent with previous reports (*Lampe, 1994*; *Sirnes et al., 2009*).

Mutations in *GJA1*, the gene encoding Cx43, have been linked to a number of diseases such as the inherited oculodentodigital dysplasia (*Paznekas et al., 2009*). We therefore asked whether PARIS could be used to probe the function of Cxs encoded by mutated Cx genes. We performed PARIS in HeLa cells, which do not express measurable levels of endogenous Cxs (Elfgang et al., 1995). As expected, no PARIS signal was elicited in receiver HeLa cells upon photoactivating the actuator cell; while in HeLa cells expressing GJA1, photoactivating the actuator cell elicited a robust fluorescence increase in the adjacent receiver cell (*Figure 3C,D*). Interestingly, expressing a Cx43 protein with either the R202H or R76H mutation—which affects Cx43 trafficking and gap junction permeability (*Shibayama et al., 2005*)—caused a significant reduction in the PARIS signal compared to cells expressing wild-type Cx43 (*Figure 3C,D*). These data indicate that PARIS can be used to probe the effects of clinically relevant mutations in gap junction proteins.

## PARIS can report the activity of functional gap junctions between cardiomyocytes

Next, we examined whether PARIS can be used to study gap junctions in a physiologically relevant system, namely cardiomyocytes (CMs). Gap junctions formed by Cx40, Cx43, and Cx45 play an important role in CMs by synchronizing their contractions and defects in these connexins have been associated with cardiovascular diseases (*Jongsma and Wilders, 2000*). Using CMs cultured from neonatal rats (*Figure 3E*), we observed that stimulating actuator CMs induced a robust fluorescence increase in receiver CMs with half-rise and half-decay times of approximately 14 and 21 s respectively, and the responses were reversibly blocked by the gap junction blocker heptanol (*Garcia-Dorado et al., 1997*) (*Figure 3F–H*). Neither the rate of spontaneous $Ca^{2+}$ transients in CMs nor the rate of cellular beating was altered by the expression or stimulation of the actuator protein (*Figure 3—figure supplement 1*), supporting the notion that expressing and activating PARIS does not affect cellular functions.

## PARIS can be used to measure cell-specific GJC in *Drosophila* brain

We then examined whether PARIS can be used to measure gap junction activity (i.e. electrical synapses) between genetically defined cell types in the brain. Using the *Drosophila* olfactory system as a model system, we first confirmed that ArchT and pHluorin can produce cell-autonomous signals in

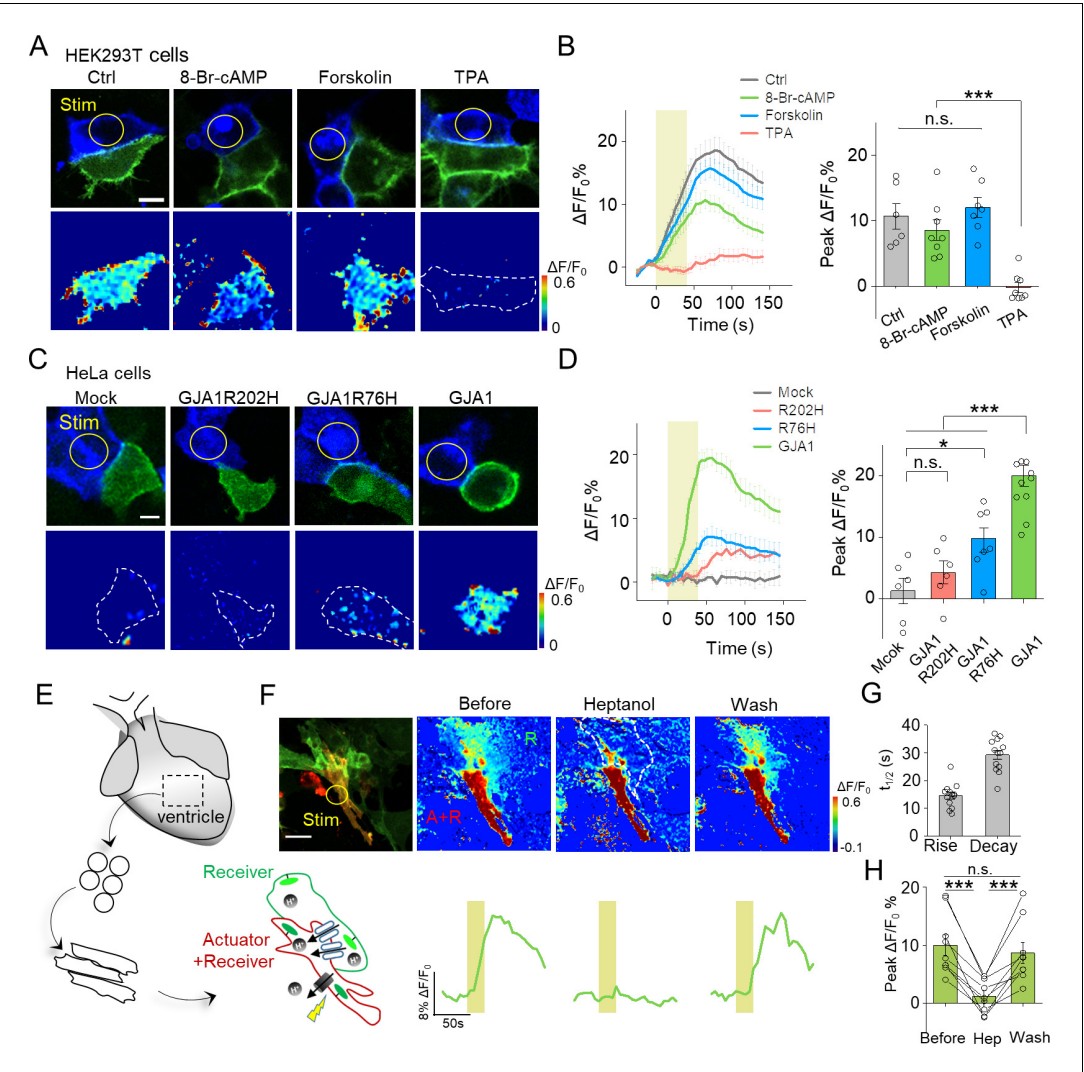

**Figure 3.** Use of PARIS to measure GJC in cultured cell lines and primary cardiomyocytes. (**A**) Top, representative images showing adjacent HEK293T cells expressing ArchT/pHluorin. Bottom, pseudocolor images of 40 s laser illumination–induced peak PARIS signals in receiver cells. Where indicated, the cells were treated with DMSO (Ctrl), 8-Br-cAMP (500 µM), Forskolin (10 µM), or TPA (340 nM) for 6 hr before PARIS measurements. (**B**) Representative traces and group analysis of PARIS signals in (**A**) (n = 6–8 cells per group). (**C**) Top, representative images showing expression of ArchT/pHluorin in HeLa cells with or without transfection of various gap junction proteins. Bottom, pseudocolor images of 40 s laser illumination–induced peak PARIS signals measured in receiver cells. (**D**) Representative traces and group analysis of PARIS signals in (**C**) (n = 6–10 cells per group). (**E**) Schematic diagram depicting the application of PARIS in cultured rat cardiomyocytes (CMs). Shown below is a corresponding confocal image of the actuator CM and receiver CM, which express ArchT and pHluorinCAAX, respectively. Note that the actuator CM expresses both ArchT and pHluorinCAAX. (**F**) Top, PARIS responses in the actuator and receiver CMs before adding heptanol, 5 min after treatment of heptanol (2 mM), and 3 min after perfusion by Tyrode solution. Shown at the left is a confocal image of the actuator and receiver CMs. Bottom, corresponding traces of the experiments shown above. Note that the light stimulus elicited a response in both CMs, but only the receiver CM was sensitive to heptanol. (**G–H**) Summary of the half-rise and half-decay times of the PARIS signals and peak $\Delta F/F_0$ for pHluorinCAAX fluorescence in receiver CMs (n = 10 cells). The scale bars represent 10 µm (**A**, **C**) or 50 µm (**F**). *p<0.05, **p<0.01, ***p<0.001, N.S., not significant (p>0.05).

DOI: https://doi.org/10.7554/eLife.43366.009

The following figure supplement is available for figure 3:

**Figure supplement 1.** Spontaneous $Ca^{2+}$ transients and beating rate in rat cardiomyocytes expressing ArchT.

DOI: https://doi.org/10.7554/eLife.43366.010

an ex vivo preparation (*Figure 4—figure supplement 1*). We expressed both the actuator and the receiver with dual binary expression systems (GH146-QF > QUAS ArchT, GH146-Gal4 > UAS pHluorinCAAX) in excitatory projection neurons (ePNs) in the fly olfactory pathway (*Stocker et al., 1997*) and measured cell autonomous PARIS signal from the antenna lobe (AL) in the isolated fly brain (i.e. in the *cis* configuration, *Figure 4—figure supplement 1A–G*). The ePN autonomous signal could be elicited repeatedly in the same sample for up to 2 hr, with no obvious loss of signal strength (*Figure 4—figure supplement 1H,I*), indicating that PARIS is stable in intact living tissue.

We then used PARIS to measure electrical synapses formed between excitatory projection neurons (ePNs) and excitatory local neurons (eLNs), both of which have dendritic arborizations in the antennal lobe (AL) (*Shang et al., 2007*) (*Figure 4—figure supplement 2*, first row). We generated a transgenic *Drosophila* line expressing the actuator (GH146-QF > QUAS ArchT) selectively in ePNs and the receiver (Kras-Gal4 >UAS pHluorinCAAX) selectively in eLNs (*Figure 4A*). Stimulating a 20 μm diameter region in the AL elicited a rapid increase in pHluorinCAAX fluorescence, with half-rise and half-decay times of approximately 12 s and 29 s, respectively (*Figure 4B,C,F,G*), consistent with previously reported electrophysiological data indicating that ePNs and eLNs are electrically coupled (*Huang et al., 2010*; *Yaksi and Wilson, 2010*). Importantly, no response was elicited in the brain when the transgenic flies were pretreated with CBX or in the brain of *ShakB²* flies, which have a mutation in their gap junction proteins (*Zhang et al., 1999*), confirming that the signal measured in the receiver neurons is indeed mediated by gap junctions (*Figure 4B,C,F,G*). Next, we divided the AL into four regions based on orientation and then scanned each region, revealing that laser illumination can induce a fluorescence increase in each region (*Figure 4D,E,H*), indicating that electrical coupling is a general property between ePNs and eLNs in the AL. In addition, we examined gap junction activity between ePNs and other cell types by pairing ePNs as the receiver cells with various actuator cells that have anatomical overlap with ePNs, including inhibitory local neurons (iLNs), glial cells, and Keyon cells (*Figure 4—figure supplement 2*, second-bottom rows). However, when activated, none of these three cell types caused a measurable PARIS signal in the receiver cells (*Figure 4—figure supplement 3*), suggesting that ePNs may not form functional gap junction connections with iLNs, glial cells, or Keyon cells.

## PARIS can be used to map functional gap junctions in distinct neuronal structures

Electrical coupling between neurons could happen through dendritic networks, axon-axonal connections or somatic contacts, which contribute the signal integration and decision of neuronal firing (*Belousov and Fontes, 2013*; *Yaksi and Wilson, 2010*). Capitalizing on the entire optical nature of PARIS, we further examined whether PARIS can be used to measure functional GJC in subcellular compartments, thereby providing spatial resolution that is not accessible to traditional methods such as electrophysiological recording or dye injection. In the *Drosophila* olfactory system, ventrally localized inhibitory projection neurons (iPNs) and ePNs form gap junctions participating odor information processing (*Wang et al., 2014*). Anatomically, both ePNs and iPNs have dendrites in the AL and axons that project to the lateral horn (LH) (*Figure 6A,B*; see also *Liang et al. (2013)*; *Parnas et al., 2013*). Thus, whether gap junctions are formed between iPNs and ePNs in the AL, LH, or both is currently unknown. To answer this question, we expressed the actuator (Mz699-Gal4 > UAS ArchT) in iPNs and the receiver (GH146-QF > QUAS pHluorinCAAX) in ePNs. We then separately illuminated ePNs-iPNs overlapping regions in the AL or LH to test the presence of functional GJC. Interestingly, stimulating the AL—but not the LH—elicited a significant increase in pHluorinCAAX fluorescence, and this response was eliminated in the presence of CBX (*Figure 5B–D*). As a control, we confirmed that CBX had no significant influence on the autonomous signal measured at either the AL or LH when both the actuator and receiver were co-expressed in iPNs (i.e. in the *cis* configuration; Mz699 >ArchT/pHluorin, *Figure 5B,C,F*). These results support the notion that iPNs and ePNs form functional gap junctions at the AL (i.e. via dendrite-dendrite contacts), but not at the LH (i.e. via axon-axon contacts).

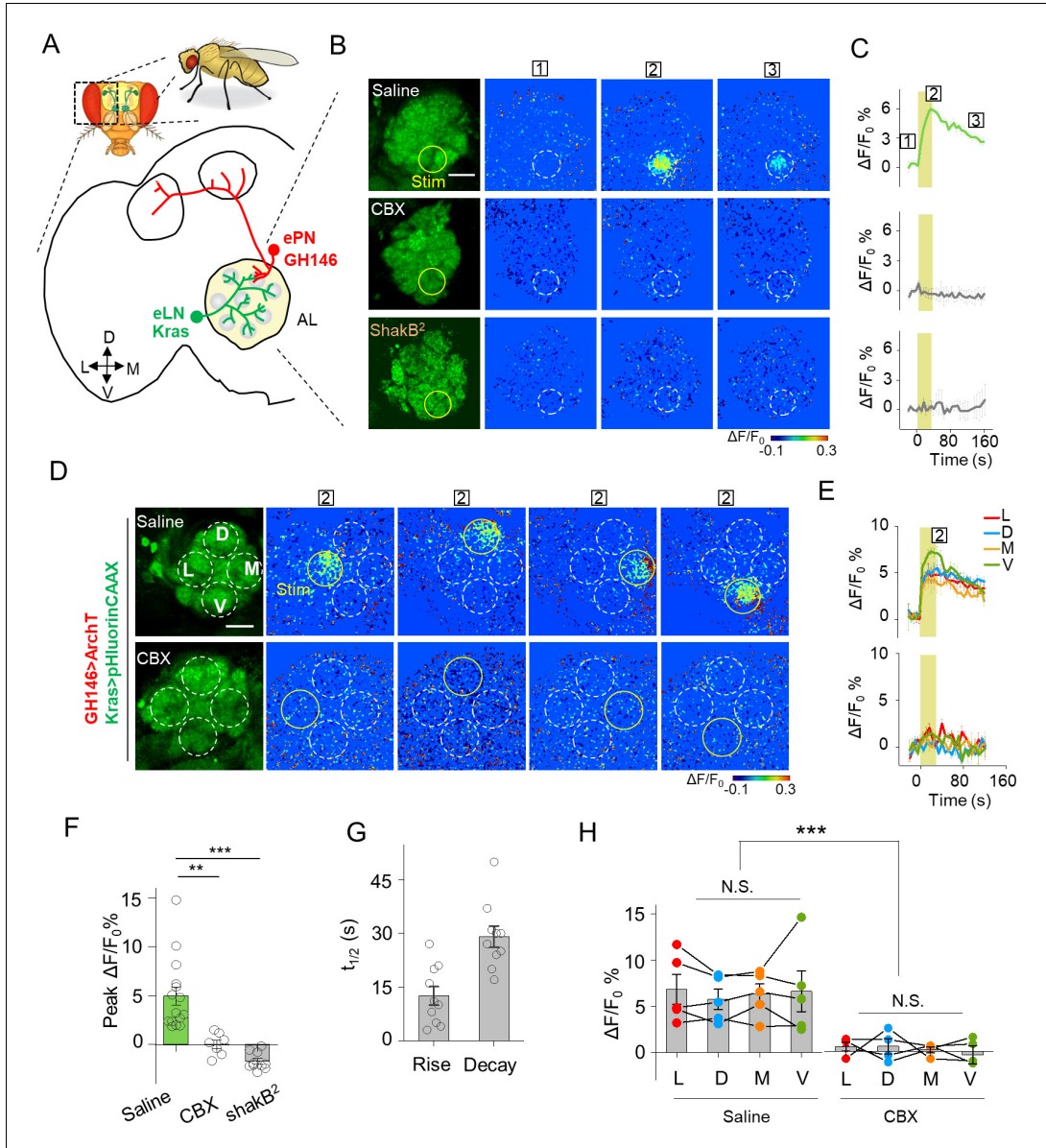

**Figure 4.** Detection of GJC between specific neurons in transgenic *Drosophila* by PARIS. (**A**) Schematic diagram depicting the anatomy of the antennal lobe (AL) in a transgenic Drosophila line in which the ePNs express ArchT and the eLNs express pHluorinCAAX (indicated in red and green, respectively). D, V, L, and M refer to dorsal, ventral, medial, and lateral, respectively. (**B, C**) Pseudocolor images (**B**) and time course (**C**) of PARIS signals in the AL of the transgenic flies shown in (**A**). Note that a 30 s pulse of 561 nm light (0.5 mW) elicited a significant PARIS signal between ePNs and eLNs (top row); in contrast, no signal was elicited when the brain was treated with 100 μM CBX (middle row, 15 min), in flies with the ShakB[2] mutation (bottom row). The boxed '1–3' above the images/traces identify stages before photostimulation, in the peak response and in the end of imaging process. (**D, E**) Pseudocolor images (**D**) and time course (**E**) showing PARIS responses of four ROIs from the lateral (**L**), dorsal (**D**), middle and ventral (**V**) part of the AL in the same transgenic *Drosophila* line in (**A**). Where indicated, the flies were treated with saline or carbenoxolone (CBX); n = 5 flies per group. (**F, G**) Summary of the peak PARIS signal (**F**) and the half-rise and half-decay times (**G**) elicited by 30 s photostimulation (n = 7–15 flies per group). (**H**) Group data for the peak PARIS response between eLNs-ePN measured in the four stimulating ROIs indicated in (**D**) (n = 5 flies per group). The scale bars 20 μm. **p<0.005, ***p<0.001, N.S., not significant (p>0.05).
DOI: https://doi.org/10.7554/eLife.43366.011

The following figure supplements are available for figure 4:

**Figure supplement 1.** Cell-autonomous PARIS signal measured in the ePNs of transgenic *Drosophila*.

*Figure 4 continued on next page*

*Figure 4 continued*

DOI: https://doi.org/10.7554/eLife.43366.012

**Figure supplement 2.** Composite confocal images of dissected fly brain expressing actuators and receivers in genetically labeled cells.

DOI: https://doi.org/10.7554/eLife.43366.013

**Figure supplement 3.** No obvious PARIS signal was detected from ePN-KC, ePN-iLN and eLN-Glia pairs.

DOI: https://doi.org/10.7554/eLife.43366.014

## Further optimization of PARIS by screening of more potent proton pumps

To further improve PARIS's performance for in vivo GJC detecting, we explored the light sensitivity of proton pumps with high-sequence homology to ArchT cloned from fungi, algae, bacteria to proteobacteria (*Figure 6A*, up right). By measuring the cell-autonomous pHluorin fluorescence increase in response to green-yellow light, we found that six candidates exhibited larger $\Delta F/F_0$ than ArchT with a fungi rhodopsin that we named Lari showed the best membrane trafficking performance and $2^6$-times the light-sensitivity of ArchT (*Figure 6A–C*, *Figure 6—figure supplement 1*). Therefore, Lari provides a more powerful actuator for in vivo application of PARIS in the future.

## Discussion

Here, we describe the development of PARIS, a new all-optical approach for detecting GJC in specific cells. We show that PARIS can be readily adapted for use in both in vitro and ex vivo preparations, including cultured cell lines, primary cardiomyocytes, and transgenic flies. We validated that this system specifically reports functional gap junctions using a variety of electrophysiological, pharmacological, and genetic approaches. By focusing on defined sets of neurons in the *Drosophila* olfactory system, we show that as the first completely genetically encoded method, PARIS can be applied to repeatedly probe electrical synapses in distinct, genetically tractable neurons with high temporal resolution (on the order of ~10 s) and high spatial resolution.

### Choice of the actuator and receiver

We initially screened three pairs of actuators/receivers, namely ArchT/pHluorin, BeCylOp/FlincG3 and CsChrimson/GCaMP6s. The latter two pairs failed to function in cis to generate receiver responses by activating the actuator (*Figure 1—figure supplement 1*). For the cGMP based pair, we have also performed the *cis* experiments in the presence of PDE inhibitor IBMX that prevented the cGMP hydrolyzation and still observed no signal; meanwhile FlincG did response to exogenous application of cGMP (data not shown). Thus, one possible explanation for the absence of the autonomous signal is that light activation of BeCylOp generated limited cGMP that could not induce FlincG3 ($EC_{50}$ = 0.89 μM) (*Bhargava et al., 2013*) response. For the pair with CsChrimson, a non-selective cation channel allows not only $Ca^{2+}$ but also other cations to pass the channel (*Klapoetke et al., 2014*), we deduce the photoactivation induced $Ca^{2+}$ influx in the CsChrimson expressing HEK293T cells was still under the detection limit of GCaMP6s. Indeed, we found CsChrimson/GCaMP6s could function in cis to generate cell autonomous signals in cultured hippocampus neurons that endogenously express voltage-gated $Ca^{2+}$ channel to allow further $Ca^{2+}$ influx (data not shown).

### Advantages of PARIS over existing techniques

First, PARIS relies solely on light and therefore is virtually non-invasive compared with existing methods including paired-recording (*Bennett et al., 1963*; *Furshpan and Potter, 1959*), dye microinjection (*Stewart, 1978*) and scrape loading (*el-Fouly et al., 1987*). In addition, given that the activation of the actuator can be specific to subcellular resolution, PARIS can provide spatial information of the functional gap junctions, as shown by our ability to functionally map gap junctions formed at dendrite-dendrite contacts in AL but not at axon-axon contacts in LH between ePNs and iPNs in the *Drosophila* olfactory system (*Figure 5*), while such resolution cannot be easily achieved by any of the previously existed method.

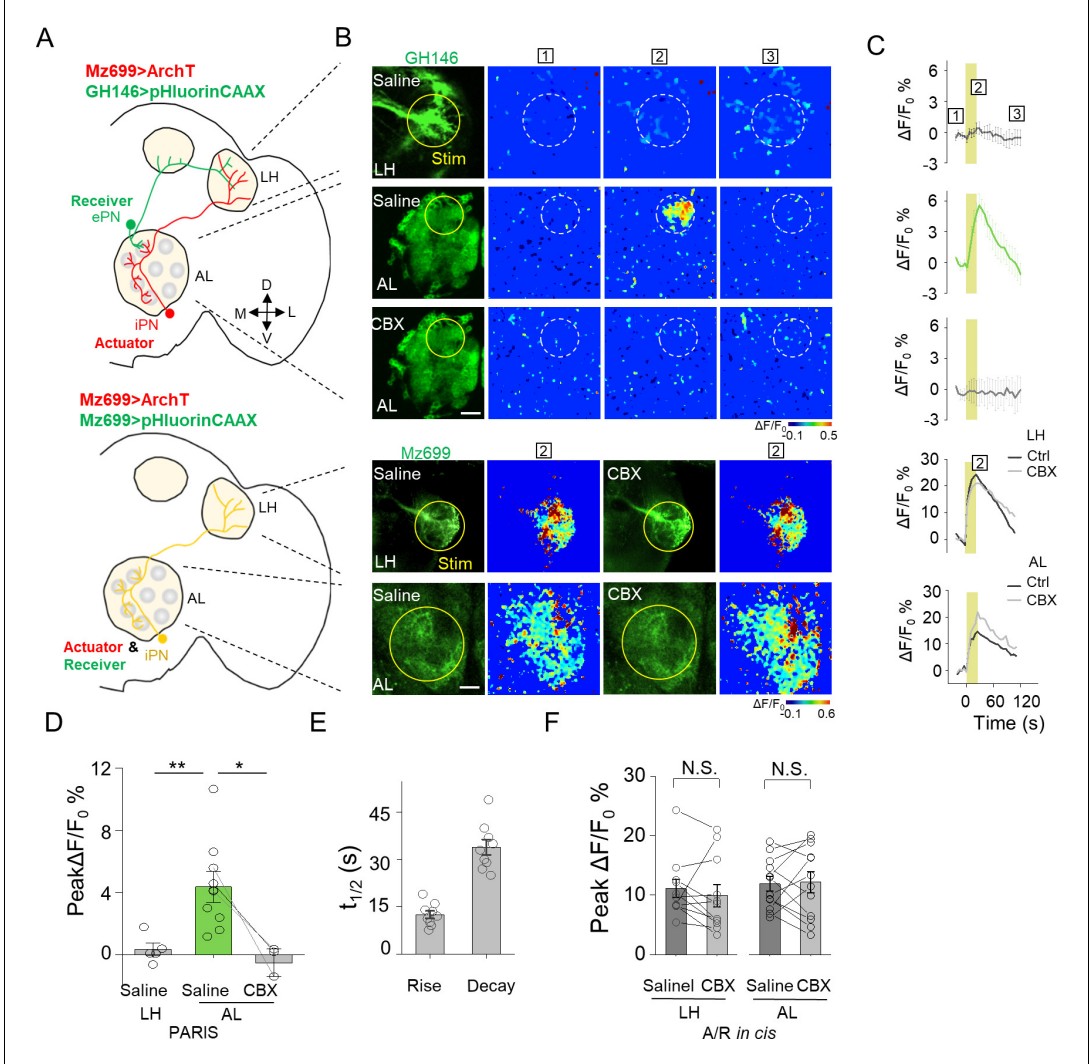

**Figure 5.** Mapping GJC at specific subcellular structures in transgenic *Drosophila* by PARIS. (A) Schematic diagrams depicting two transgenic *Drosophila* lines in which the ePNs express pHluorinCAAX and iPNs express ArchT (top, shown in green and red), or only the iLNs co-expressing ArchT and pHluorinCAAX (bottom, shown in yellow) in the olfactory pathway are indicated. (B, C) Pseudocolor images (B) and time course (C) of PARIS signals in the AL and LH regions of the transgenic flies shown in (A). Note that in flies in which the ePNs and iPNs express the receiver and actuator, respectively, a 20 s pulse of yellow light elicited a significant PARIS signal in the AL, but not in the LH. Moreover, the signal induced in the AL was inhibited by 100 μM CBX. CBX had no effect in flies in which the actuator and receiver were co-expressed in the same iPNs (i.e., in the cis configuration). (D–F) Group data for the peak PARIS signals (D), the half-rise and half-decay times measured in the indicated conditions (E) and the iPN autonomous responses (n = 3–11 flies per group). The scale bars in (B) represent 20 μm. *p<0.05, N.S., not significant (p>0.05).

DOI: https://doi.org/10.7554/eLife.43366.015

The following figure supplement is available for figure 5:

**Figure supplement 1.** Composite confocal images of dissected fly brain expressing actuators and receivers in iPNs and ePNs.
DOI: https://doi.org/10.7554/eLife.43366.016

With respect to those relatively non-invasive methods which rely on the diffusion of small fluorescent dyes across gap junctions such as FRAP (*Wade et al., 1986*) and LAMP (*Dakin et al., 2005*), a significant advantage of PARIS is that it is fully reversible and does not require the delivery of exogenous dyes. PARIS may serve as a robust tool for screening existing and newly developed gap junction blockers and/or modulators, including clinically relevant compounds such as inhibitors of PKC signaling. In addition, PARIS could possibly be applied to study the dynamic regulation of gap junctions in vivo, such as the formation and break of gap junction connections during brain development (*Arumugam et al., 2005*).

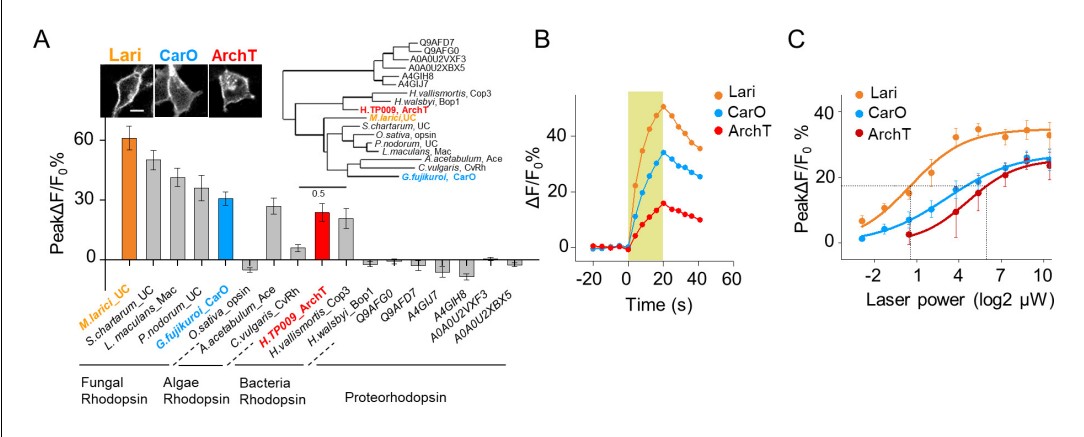

**Figure 6.** Further optimization of PARIS by screening of more potent proton actuators. (**A**) Screening for high-efficiency proton actuators. Actuators were fused with BFP at the C-terminus and co-expressed with pHluorinCAAX in HEK293T cells. Top left, membrane trafficking performance of two candidates besides ArchT; top right, phylogenic tree of screened rodopsins. The tree was built using PhyML (**Guindon and Gascuel, 2003**); bottom, cell-autonomous pHluorin signals of all the potential proton pumps under 20 s photostimulation (n = 5–13 cells per group). (**B,C**) Exemplary responses and grouped cell-autonomous peak $\Delta F/F_0$ % of ArchT, Lari and CarO under increased laser power (20 s stimulation time). The data were fit to a single Hill function (solid lines, n = 10 cells for each data point). The scale bar in (**A**) represents 10 μm. .

DOI: https://doi.org/10.7554/eLife.43366.017

The following figure supplement is available for figure 6:

**Figure supplement 1.** Membrane traffic performance of two proton-pumps compared with ArchT.

DOI: https://doi.org/10.7554/eLife.43366.018

Moreover, the actuator and receiver proteins in PARIS are both genetically encoded. Recently, several innovative hybrid approaches were developed to monitor gap junctions (**Kang and Baker, 2016**; **Qiao and Sanes, 2015**; **Tian et al., 2012**). Using a channel/transporter/enzyme-dependent step for the transfer of small molecules, these approaches can in principle achieve genetic specificity in terms of targeting defined cells. In practice, however, these methods require the addition of an exogenous substrate (**Qiao and Sanes, 2015**; **Tian et al., 2012**) or a patch electrode to establish an electrochemical gradient between connected cells (**Kang and Baker, 2016**), thereby limiting their application, particularly in vivo. In contrast, all the components in PARIS are genetically encoded by relatively small genes, vastly increasing the range of cell types in which they can be selectively expressed. For example, we show that the PARIS proteins can be introduced using transfection (**Figures 1–3**), virus-mediated expression (**Figure 3**), and ex vivo transgenic labeling (**Figure 4 and 5**). Given that similar transgenic tools are available for higher organisms, including mammals, PARIS can easily be adapted to other preparations and animal models, including the highly complex mammalian nervous system. In mammalian systems and model organisms in which transgenic techniques are currently unavailable or impractical, recombinant viruses such as lentiviruses, retroviruses, and adeno-associated viruses can be conveniently packaged with these genetic components and used to infect the appropriate target cells.

## Limitations of PARIS

In most animal cells, intracellular pH is believed to be tightly regulated for optimal biochemical reactions; thus, even a small change in intracellular pH is rapidly restored to a set point by buffers and proton exchangers/transporters located on the membrane of cells or intracellular organelles (**Hoffmann and Simonsen, 1989**). This robust system for maintaining pH homeostasis enabled us to repeatedly elicit a PARIS signal in both cultured cells and transgenic animals. One caveat to our approach may be the ability of pH to regulate gap junction activity during PARIS. The uncoupling effect of acidic intracellular pH on GJC has long been described across different Cx-consisted gap junctions in vertebrates (**Peracchia, 2004**; **Turin and Warner, 1977**) and different innexin-consisted gap junctions in invertebrates (**Giaume et al., 1980**; **Obaid et al., 1983**; **Stergiopoulos et al., 1999**), while alkalization was reported to increase the junctional conductance and the total number

of opened channels in gap junction plagues (*Palacios-Prado et al., 2010*). For the mostly wide expressed Cx43 channels, it has a pKa of ~6.7 in oocytes and fully closed when pH is under 6.4 while fully open when pH is above 7.2 (*Stergiopoulos et al., 1999*), which enables PARIS measurement to reveal the GJC mediated by Cx43. However, there is one type of gap junctions that has been reported to be sensitive to alkalization—Cx36 consisted gap junctions (*González-Nieto et al., 2008*). Based on the reported pH-conductance curve, the junctional conductance decreased to 50% when pH increased 0.8 unit from 7.2 to 8. As shown in *Figure 1—figure supplements 4* and 0.1-unit pH increase from 7.35 to 7.45 was enough to induce PARIS signal. So PARIS is still possible in reporting Cx36 consisted GJC under proper activation of the actuator. Even though we conclude that PARIS induced pH fluctuation is controllable and transient, one should still be cautious to the possible modification towards gap junctions as well as cell physiology. For the long-time measurements, either reduce the power or shorten the time of laser illumination, meanwhile increase the interval between each measurement should be helpful to decrease the pH influence. An even more sensitive pH indicator could help to minimize the pH influence as well.

As we have demonstrated, PARIS is powerful as a genetic tool to map gap junction connections between targeted neurons in the complex central nervous system (*Figure 4 and 5*). To map unknown gap junctions, firstly we need anatomic information about the two group of cells that we concern to make sure they are spatially contacted, which could be achieved by immunostaining or EM. As the intensive efforts have been or being made to create whole brain connectomes from *C. elegans* (*Jarrell et al., 2012*), *Drosophila* (FlyEM project, Janelia Campus Research) to zebra fish (*Hildebrand et al., 2017*) and mice (Allen Mouse Brain Connectivity Atlas), PARIS could utilize these information and databases and possibly help creating electrical connectome.

PARIS requires the exclusive expression of the actuator and receiver in different cells. Such genetic tools specifically labeling two distinct groups of cells from different cell types or two subgroups from one cell type, might not be accessible in mammals. However, this limitation can be overcome by the intersectional non-overlapping expression of the actuator and receiver, for example using flp-out system. By fusing the receiver with a flippase, the frt sequence flanked actuator would not express in the presence of the receiver. Meanwhile the receiver and the actuator can both designed to be turned on in a cre-dependent manner. This design could make PARIS more versatile in detecting GJC between specific cells labeled by cre-lines without the contamination of the autonomous signal.

Lastly, to protect against false negatives of PARIS, a control experiment in the cis configuration is recommended in different customized preparations and context to ensure the function of the actuators and help to optimize the expression level of actuators/receivers as well as the photostimulation parameters accordingly; Meanwhile, PARIS signals alone from cells connected by potential unknown gap junctions should not be interpreted as definitive without confirmation from pharmacology, genetic interventions or a complementary method.

## Future perspectives

Future refinements to PARIS include the use of the new actuators we have screened combining a receiver with higher pH sensitivity, thereby increasing both the signal-to-noise ratio and temporal resolution, allowing for an even wider range of in vitro and in vivo applications. Finally, the use of additional spectrally non-overlapping pairs of proton-related actuators and receivers, as well as developing actuator-receiver pairs that transport and detect other gap junction–permeable small molecules, may provide the opportunity to detect gap junctions between multiple cell types and/or structures.

## Materials and methods

**Key resources table**

| Reagent type (species) or resource | Designation | Source or reference | Identifiers | Additional information |
|---|---|---|---|---|

*Continued on next page*

*Continued*

| Reagent type (species) or resource | Designation | Source or reference | Identifiers | Additional information |
|---|---|---|---|---|
| Genetic reagent (*Drosophila melanogaster*) | GH146-Gal4 | Liqun Luo lab | RRID:BDSC_30026 | |
| Genetic reagent (*D. melanogaster*) | GH146-QF | Liqun Luo lab | RRID: BDSC_30015 | |
| Genetic reagent (*D. melanogaster*) | GH298-Gal4 | Liqun Luo lab | RRID: BDSC_37294 | |
| Genetic reagent (*D. melanogaster*) | Krasavietz-Gal4 | Donggen Luo lab | flybaseID#_ FBti0027494 | |
| Genetic reagent (*D. melanogaster*) | ShakB[2] | Donggen Luo lab | flybaseID#_ FBal0015575 | |
| Genetic reagent (*D. melanogaster*) | Mz699-Gal4 | Liqun Luo lab | flybaseID#_ FBti0007260 | |
| Genetic reagent (*D. melanogaster*) | Repo-Gal4 | Yi Rao lab | RRID:BDSC_7415 | |
| Genetic reagent (*D. melanogaster*) | MB247-Gal4 | Yi Rao lab | RRID:BDSC_64306 | |
| Genetic reagent (*D. melanogaster*) | UAS-pHluorinCAAX | This study | | |
| Genetic reagent (*D. melanogaster*) | QUAS-pHluorinCAAX | This study | | |
| Genetic reagent (*D. melanogaster*) | UAS-ArchT-P2A-mRuby3CAAX | This study | | |
| Genetic reagent (*D. melanogaster*) | QUAS-ArchT-P2A-mRuby3CAAX | This study | | |
| Cell line (Human) | HEK293T | ATCC | Cat#CRL-3216; RRID:CVCL_0063 | |
| Cell line (Human) | HeLa | ATCC | Cat#CCL-2; RRID:CVCL_0030 | |
| Chemical compound,drug | CBX | Sigma-Aldrich | Cat#C4790 | |
| Chemical compound,drug | Heptanol | J and K Scientific | Cat#415422 | |
| Chemical compound,drug | Forskolin | TargetMol | Cat#T2939 | |
| Chemical compound,drug | 8-Br-cAMP | Sigma-Aldrich | Cat#B7880 | |
| Chemical compound,drug | TPA | Sigma-Aldrich | Cat#P1585 | |
| Chemical compound,drug | Calcein-AM | AAT Bioquest | Cat#22002 | |
| Chemical compound,drug | Fluo-8-AM | AAT Bioquest | Cat#21082 | |
| Recombinant DNA reagent | pcDNA3.1 vector | Michael Lin lab | Addgene: 52519 | |
| Recombinant DNA reagent | pcDNA3.1-Lari-BFP2 | This study | | |
| Recombinant DNA reagent | pJFRC28 | (*Pfeiffer et al., 2012*) | Addgene: 36431 | |
| Recombinant DNA reagent | pJFRC28-10xUAS-pHluorinCAAX-p10 | This study | | |
| Recombinant DNA reagent | pJFRC28-5xQUAS pHluorinCAAX-p10 | This study | | |

*Continued on next page*

*Continued*

| Reagent type (species) or resource | Designation | Source or reference | Identifiers | Additional information |
|---|---|---|---|---|
| Recombinant DNA reagent | pJFRC28-10xUAS–ArchT-P2A-mRuby 3CAAX-p10 | This study | | |
| Recombinant DNA reagent | pJFRC28-5xQUAS -ArchT-P2A-mRuby 3CAAX-p10 | This study | | |
| Sequence-based reagent | human ORFeome 8.1 | Center for Cancer Systems Biology | http://horfdb.dfci .harvard.edu/ | Full-length human cDNAs |
| Software, algorithm | ImageJ | NIH | https://imagej.nih.gov/ij/; RRID:SCR_003070 | |
| Software, algorithm | Origin 9.1 | OriginLab | https://www.originlab.com/ | |
| Software, algorithm | MATLAB | MathWorks | https://www.mathworks.com/ products/matlab.html; RRID:SCR_001622 | |
| Other | Inverted confocal microscope | Nikon | Ti-E A1 | |

## Molecular cloning

All plasmids were constructed using the Gibson assembly (*Gibson et al., 2009*) method. In brief, plasmid vectors and inserts were amplified by PCR using ~30 bp overlapping primers. The fragments were assembled using T5 exonuclease, Phusion DNA polymerase, and Taq DNA ligase (New England Biolabs). All sequences were verified using Sanger sequencing in our in-house facility (sequencing platform in the School of Life Sciences of the Peking University). For cultured cell expression experiments, genes were cloned into the pcDNA3.1 vector unless otherwise noted. ArchT was amplified from pAAV-CAG-ArchT-GFP (*Han et al., 2011*), codon-optimized rhodopsin genes from different species were synthesized by Qinglan Biotec then fused at the C-terminus with BFP2, a trafficking sequence (TS), and an ER export sequence (ERex), producing Actuator-BFP2. In addition, ArchT was linked directly with a TS and ERex and then fused with mRuby3, yielding pcDNA3.1-ArchT-P2A-mRuby3. mTagBFP was fused to the N-terminus of pHluorinCAAX to generate pcDNA3.1-mTagBFP-pHluorinCAAX. The *GJA1* gene was amplified from a cDNA library (hORFeome database 8.1) and fused with RFP (pHuji) (*Shen et al., 2014*) via a P2A linker to generate pHuji-P2A-GJA1, which was then cloned into the N3 expression vector. Mutations in ArchT and GJA1 were introduced using overlapping PCR with primers containing the mutations of interest. To generate transgenic *Drosophila* lines, the following four plasmids were used: ArchT (linked with mRuby3CAAX, HA tag and Flag tag: ArchT-HA-TS-ERex-P2A-mRuby3-Flag-CAAX) and pHluorinCAAX were cloned into pJFRC28-10xUAS vector (*Pfeiffer et al., 2012*)/pJFRC28-5xQUAS vector (made by replacing 10xUAS with 5XQUAS in pJFRC28) respectively, yielding UAS/QUAS-ArchT and UAS/QUAS-pHluorin transgenic flies.

## Cell culture and transfection

HEK293T cells and HeLa cells were purchased and certified from ATCC (ATCC, Gaithersburg, MD). The cells were negative for mycoplasma. HEK293T cells and HeLa cells were cultured in DMEM containing 10% (v/v) FBS and 1% penicillin-streptomycin (all from Gibco) at 37°C in humidified air containing 5% $CO_2$. For transfection, cells were plated at 50% confluence on 12 mm glass coverslips in 24-well plates; 12–24 hr after plating, the cells were transfected using polyethylenimine (PEI), with a typical ratio of 1 µg DNA: 3 µg PEI per well; 6 hr after transfection (or 2 hr after transfection for electrophysiological experiments), the culture medium was replaced with fresh medium and the cells were incubated for an additional 18–24 hr prior to imaging or electrophysiological recording. For PARIS transfection, the ArchT-BFP2 and pHluorinCAAX constructs were transfected in separate wells; 6 hr after transfection, the cells were dissociated, mixed by pipette, combined into a single well, and incubated for 24 hr prior to imaging. Alternatively, in some experiments, we used sequential transfection, in which the cells were first transfected with the pHluorinCAAX construct; 6 hr later,

the medium was changed and the cells were transfected with the ArchT-BFP2 construct. The medium was changed 6 hr later, and the cells were incubated for 24 hr prior to imaging.

For PARIS measurements in HeLa cells, 0.5 μg pHluorinCAAX was mixed with 0.5 μg pHuji-P2A-GJA1 (or the R202H or R76H mutant version) and transfected into the cells; 10 hr later, the medium was replaced with new medium and the cells were transfected with 0.5 μg ArchT-BFP2 mixed with 0.5 μg pHuji-P2A-GJA1 (or the R202H or R76H mutant version). The medium was replaced 10 hr later, and the cells were incubated for an additional 24 hr prior to imaging. pH calibration in HEK293T cells.

HEK293T cells were co-transfected with the mTagBFP-pHluorinCAAX and ArchT-P2A-mRuby3 plasmids (each at 0.5 μg per well in a DNA:PEI ratio of 1:3) or mTagBFP-pHluorinCAAX alone (1 μg per well) 24 hr prior to imaging. Cover slips with cells attached were immersed in Tyrode's solution containing (in mM): 150 NaCl, 4 KCl, 2 $MgCl_2$, 2 $CaCl_2$, 10 HEPES, and 10 glucose (pH 7.4) and then pre-treated with 10 μM Nigericin for 5 min, after which the calibration buffers were perfused into the chamber. After Nigericin was added, GFP and BFP channels were recorded simultaneously at 5 s intervals (500 ms/frame, 512 × 512 pixels) using a Nikon A1 confocal microscope. Calibration buffers (containing 10 μM Nigericin) at pH 7, 7.5, 8, 8.5, and 9.5 contained (in mM) 120 KCl, 30 NaCl, 2 $MgCl_2$, 10 Glucose, and 10 HEPES; for pH 5.5 and 6.5 buffers, HEPES was replaced with 10 mM MES.

## Electrophysiology

Patch-clamp recordings were performed using an Olympus IX81 upright microscope equipped with a 40x/0.95 NA objective; images were acquired using Micro-Manager (https://micro-manager.org/). Laser light was delivered via a Sutter DG-4 equipped with a xenon lamp. Cultured HEK293T cells were bathed in Tyrode's solution. Actuator HEK293T cells were identified by blue fluorescence (excitation filter: 350/50 nm; emission filter: 448/60 nm). ArchT was activated using the same light source filtered through a 560/25 nm Sutter VF5 filter. Light intensity was adjusted using the Sutter DG-4 and was calculated by measuring the light power transmitted through the objective using a SANWA laser power meter LP1). Recording electrodes (with a tip resistance of 3–6 MΩ when filled with internal solution) were fabricated using a Sutter P-97 electrode puller and controlled using Sutter MP-225 micromanipulators. The recording electrodes were filled with an internal solution containing (in mM): 130 K-gluconate, 10 KCl, 2 $MgCl_2$, 2.5 Mg-ATP, 0.25 Na-GTP, 10 HEPES, and 0.4 EGTA (pH adjusted to 7.4 with KOH; osmolarity adjusted to 300 mOsm with sucrose); where indicated, 100 μM carbenoxolone (CBX) (Sigma-Aldrich) was applied by perfusion. The recording signal was amplified and digitized using a HEKA EPC10 double patch-clamp amplifier and collected using Patch Master. Currents were smoothed using a 20 ms moving average in order to minimize 50 Hz AC noise. For simultaneous optical and electrophysiology recordings, the Sutter DG-4 light source was triggered by the HEKA EPC10 amplifier in order to synchronize the electrophysiological recording with the light simulation. All recordings were performed at room temperature.

## FRAP measurements in HEK293T cells

A 1 mM stock solution of Calcein-AM was added to the culture medium to a final concentration of 1 μM. The cells were then incubated for 10 min before washing 3 times with 1 ml Tyrode's solution. The coverslips containing the attached cells were then placed in a chamber containing Tyrode's solution and imaged using a Nikon A1 confocal microscope. A typical FRAP experiment was performed using 488 nm imaging for measuring the baseline (1 s/frame, 5 s interval, five frames, 512 × 512 pixels), 405 nm bleaching (12 s, ROI ~5 μm in diameter within the cell), and 488 nm imaging for 2 min in order to track Calcein recovery.

## Preparation, culture, transfection, and infection of primary cardiomyocytes

Cardiomyocytes (CMs) were enzymatically dissociated from the ventricles of neonatal (P0) Sprague-Dawley rats, and 0.5–1 × $10^5$ CMs per well were seeded on 12 mm glass coverslips coated with poly-D-lysine (Sigma-Aldrich) in 24-well plates and grown in DMEM containing 10% FBS and 1% penicillin-streptomycin (all from Gibco) at 37°C in humidified air containing 5% $CO_2$. Forty-eight hours after plating, CMs were simultaneously transfected with the ArchT-BFP2 plasmids using

Lipofectamine 3000 (Invitrogen) and infected with an adenovirus carrying pHluorinCAAX under CMV promoter (Vigene Biosciences). In brief, before transfection/infection, the medium in each well was replaced with 500 µl Opti-MEM. DNA (1 µg) and Lipofectamine 3000 (1.5 µl) were diluted in 100 µl Opti-MEM and incubated for 15 min at room temperature before additional to each well. At the same time, 1 µl of adeno-associated virus carrying the pHluorinCAAX ($3 \times 10^{10}$ pfu/ml) was added into the same wells. Eight hours later, the Opti-MEM was replaced with standard DMEM medium, and the CMs were incubated for an additional 24 hr prior to imaging.

## Measurements of beating rate and Ca$^{2+}$ transients in CMs

CMs were transfected with ArchT-BFP2 plasmids using Lipofectamine 3000 as described above; 24 hr after transfection, CMs expressing ArchT-BFP2 were loaded with 1 µM Fluo-8 AM (AAT Bioquest) for 20 min, washed with 1 ml Tyrode's solution for three times. CMs were imaged with 488 nm excitation in Tyrode's solution at room temperature using a Nikon A1 confocal microscope for 1 min to record Ca$^{2+}$ transients, followed by stimulation with 561 nm for 20 s (5 trials at 2 min intervals). After stimulation, another 1 min of time-lapse imaging was performed using 488 nm excitation to record Ca$^{2+}$ transients. Ca$^{2+}$ transients recorded before and after light stimulation were counted using ImageJ analysis of the green channel. All images were generated at a rate of 100 ms/frame. Beating rate was measured using ImageJ analysis of the white-field images.

## Fly strains

*Drosophila* stocks were raised at 25°C on standard corn meal-agar-molasses medium. GH298-Gal4, Mz699-Gal4 (III) (*Ito et al., 1997*), and GH146-Gal4 (II) (*Stocker et al., 1997*) and GH146-QF (*Potter et al., 2010*) were kindly provided by Dr. Liqun Luo. Krasavietz-Gal4 (*Dubnau et al., 2003*) and ShakB$^2$ (*Zhang et al., 1999*) strains were gifts from Dr. Donggen Luo which has been verified by genotyping and sequencing the mutated site. Repo-Gal4 and MB247-Gal4 strains were gifts from Dr. Yi Rao. UAS/QUAS-ArchT and UAS/QUAS-pHluorinCAAX transgenic flies were generated at the Core Facility of *Drosophila* Resource and Technology, Institute of Biochemistry and Cell Biology, Chinese Academy of Sciences. All the transgenic flies have been genotype verified by sequencing in our in-house facility (sequencing platform in the School of Life Sciences of the Peking University).

## Ex vivo fly brain preparations for PARIS imaging

The entire brain from adult flies within 2 weeks after eclosion (no gender preference) was dissected using fine forceps into Ca$^{2+}$-free adult-like hemolymph (ALH) at RT containing (in mM): 108 NaCl, 5 KCl, 5 HEPES, five trehalose, five sucrose, 26 NaHCO$_3$, 1 NaH$_2$PO$_4$, 2 CaCl$_2$, and 1–2 MgCl$_2$ (275 mOsm) (*Wang et al., 2003*). The brain was then transferred to a glass-bottom chamber containing ALH for confocal imaging. The brain was held in place using a custom-made platinum frame and positioned with the anterior surface of the brain toward the objective (for imaging and stimulation of the antennal lobe), or with the posterior surface toward the objective (for imaging and stimulation of the lateral horn).

## Fluorescence imaging and light stimulation

Imaging and light stimulation were performed at RT using an inverted Nikon A1 confocal microscope equipped with a Nikon sapphire laser and either a 40x/1.35 NA oil objective (for HEK293T and HeLa cells) or a 20x/0.75 NA air objective (for CMs and fly brains). During imaging, the cells (HEK293T cells, HeLa cells, or CMs) were immersed in Tyrode's solution. Cells expressing the actuator (ArchT-BFP2) or receiver (pHluorinCAAX) were identified by the presence of blue or green fluorescence respectively. ArchT-BFP2 was excited at 405 nm and visualized using a 450/50 nm filter; pHluorin was excited at 488 nm and visualized using a GaSP photomultiplier after passing through a 525/50 nm filter. Blue fluorescent cells in adjacent to green cells were selected to conduct photostimulation with a ROI of 10–20 µm in diameter. ArchT was photostimulated using a 561 nm scanning laser at 0.5 mW for experiments using cell lines (except for the power-dependent measurements) and 0.1 mW for experiments using CMs. Typically, cells were first imaged using 488 nm excitation (256 × 256 pixels, 500 ms/frame, 2–5 s intervals) to obtain a baseline fluorescence measurement. After obtaining a baseline, pulses of 561 nm laser light were applied to activate the ArchT (4 s/pulse, 5–10

pulses with no delay), intertwined with 488 nm imaging of the receiver. The 488 nm imaging was continued (2–5 s intervals) for 1–2 min after the 561 nm stimulation in order to record the fluorescence recovery of the receiver. For experiments using gap junction blockers and modulators, 100 µM CBX (Sigma-Aldrich), 2 mM heptanol (J and K Scientific), 500 µM 8-Br-cAMP (Sigma-Aldrich), 500 µM forskolin (TargetMol), or 340 nM TPA (Sigma-Aldrich) was applied by adding a 1000x stock solution to the chamber. For CM experiments, heptanol was washed out of the chamber for 3 min by perfusion with Tyrode's solution.

Fly brains were stimulated and imaged in ALH using the same laser configuration described above at RT. Genotypes of samples were verified by both the presence and the pattern of green (pHluorin) or red (ArchT) fluorescence. The antennal lobe (AL) and lateral horn (LH) were identified by the green fluorescence of pHluorin and were stimulated with 0.5 mW laser intensity using an ROI 20–30 µm in diameter. Ctrl brain and *shakeB* mutant brain were photostimulated and imaged in ALH after dissection, brain in CBX group were immersed in ALH containing 100 µM CBX for 15 min before photostimulation and imaging.

## Data analysis

Time series images were analyzed using Nikon NIS and ImageJ with a stack stablizing plugin (http://www.cs.cmu.edu/~kangli/code/Image_Stabilizer.html). A mean background value obtained from regions away from the pHluorin was subtracted in order to correct for fluorescence intensity (F). For each pixel, we calculated the normalized change in fluorescence intensity ($\Delta F/F_0$), where $F_0$ is the baseline fluorescence averaged from five frames obtained prior to light stimulation. $\Delta F/F_0$ over time was further processed using Origin 9.1 (OriginLab). $\Delta F/F_0$ images were processed with MATLAB (MathWorks) using custom-written scripts in order to produce pseudocolor images which is provided as *Source code 1*.

## Data presentation and statistical analysis

All summary data are reported as the mean ±s.e.m. The raw data of each cell or brain sample are presented in the graphs and the sample size are indicated in the legends. All data analyses were performed using Origin 9.1 (OriginLab). Differences between groups were tested using the Student's unpaired or paired *t*-test, and differences with $p < 0.05$ (two-sided) were considered statistically significant. For *Figure 4H*, a two-way ANOVA with Tukey post-hoc test was used.

## Acknowledgements

We thank L Luo, Y Rao, D Luo and the Bloomington Drosophila Stock Center for sharing their *Drosophila* strains. We thank Core Facility of Drosophila Resource and Technology of Shanghai Institute of Biochemistry and Cell Biology, CAS for help in creating transgenic flies. We are also grateful to the members of the Y Li lab who provided feedback on the manuscript. This work was supported by the National Basic Research Program of China (973 Program; grant 2015CB856402 to YL and 2016YFA0500401 to SQW), the General Program of National Natural Science Foundation of China (projects 31671118 and 31371442, to YL and 31630035 to SQW), the Junior Thousand Talents Program of China (to YL), and 2018 Beijing Brain Initiative (Z181100001518004 to YL).

## Additional information

### Funding

| Funder | Grant reference number | Author |
| --- | --- | --- |
| National Natural Science Foundation of China | 31371442 | Yulong Li |
| National Natural Science Foundation of China | 31671118 | Yulong Li |
| National Natural Science Foundation of China | 31630035 | Shi-Qiang Wang |

| Ministry of Science and Technology of the People's Republic of China | 2015CB856402 | Yulong Li |
| Ministry of Science and Technology of the People's Republic of China | 2016YFA0500401 | Shi-Qiang Wang |
| Beijing Brain Initiative | Z181100001518004 | Yulong Li |

The funders had no role in study design, data collection and interpretation, or the decision to submit the work for publication.

### Author contributions
Ling Wu, Data curation, Formal analysis, Validation, Investigation, Methodology, Writing—original draft, Writing—review and editing; Ao Dong, Data curation, Validation, Investigation, Writing—review and editing; Liting Dong, Data curation, Formal analysis, Validation, Investigation, Writing—review and editing; Shi-Qiang Wang, Supervision, Funding acquisition; Yulong Li, Conceptualization, Supervision, Funding acquisition, Writing—original draft, Writing—review and editing

### Author ORCIDs
Ling Wu http://orcid.org/0000-0003-3921-5626
Ao Dong https://orcid.org/0000-0002-2821-9528
Liting Dong http://orcid.org/0000-0001-8396-374X
Yulong Li https://orcid.org/0000-0002-9166-9919

### Decision letter and Author response
Decision letter https://doi.org/10.7554/eLife.43366.023
Author response https://doi.org/10.7554/eLife.43366.024

## Additional files
### Supplementary files
• Source code 1. pseudocolor.
DOI: https://doi.org/10.7554/eLife.43366.019

• Transparent reporting form
DOI: https://doi.org/10.7554/eLife.43366.020

### Data availability
All data generated or analysed during this study are included in the manuscript and supporting files.

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
