## [Decision Letter]

Thank you for submitting your article "PARIS, an optogenetic method for functionally mapping gap junctions" for consideration by *eLife*. Your article has been reviewed by three peer reviewers, and the evaluation has been overseen by a Reviewing Editor and Eve Marder as the Senior Editor. The following individuals involved in review of your submission have agreed to reveal their identity: David J. Schulz (Reviewer #1); Alberto Pereda (Reviewer #2); James Jeanne (Reviewer #3).

The reviewers have discussed the reviews with one another and the Reviewing Editor has drafted this decision to help you prepare a revised submission.

During the consultation session, the three reviewers were in agreement that no new experiments are required, but that there are issues that will require editorial changes and some rewriting. I am taking the unusual tactic of including all three complete reviews so that you have the benefit of the context for the concerns as you prepare your revisions.

*Reviewer #1:*

The authors present a novel optical method to visualize gap junction connections (PARIS). Strengths of this new method include non-invasiveness, subcellular resolution, and the ability to apply this through transgenic means that allow for targeting to specific defined cell types.

Overall, I find the work novel and innovative and potentially interesting to a growing field of researchers focused on these underappreciated cellular connections. The manuscript itself is very well crafted – and the figures of very high quality throughout. In many ways this was a pleasure to read. Of particular note is the authors' care to appropriately discuss the benefits and limitations of the new technique.

If I have one major concern with the utility of PARIS, it is that it seems to be at this point a largely qualitative tool. The authors have demonstrated that functional coupling can be identified with PARIS. What would truly be exceptional is if this could be used to measure not only the presence of coupling, but the relative strength of coupling as well. To do so, the authors would need to measure coupling strength with a traditional approach (e.g. current clamp or voltage clamp), measure with PARIS in the same cells, and correlate this across a number of cell pairs with varying coupling strengths. While I do not think this is required for this study to be ready for publication, it would greatly enhance the utility of this approach.

My only other concern, which is touched on by the authors, is that PARIS will manipulate intracellular pH – and invoke homeostatic mechanisms to maintain cellular pH. If this occurs, there may be physiological questions that alter cellular properties as a result. This may limit the utility of PARIS for longitudinal studies where multiple measurements of coupling are taken over the time course of some manipulation.

*Reviewer #2:*

This paper 'PARIS, an optogenetic method for functionally mapping gap junctions' by Wu at al. describes a novel approach to explore gap junctional communication. Taking advantage of genetics the method pairs an actuator (a light gated outward proton pump ArchT) in one cell with a receiver (the pH sensitive GFP pHluorin) in an adjacent cell, a combination that has the ability of to detect gap junctional communication between these two cells. The development of this method represents a significant contribution to the current toolbox of techniques available to study intercellular communication via gap junctions because:

a) It allows repeated measurements of gap junctional coupling.b) It is cell-specific.c) Although not comparable with electrophysiological approaches, it has an acceptable temporal resolution, better than that of comparable methods.d) More importantly, it has exquisite spatial resolution, a property without parallel in any other method.e) Combined, these properties make this new method ideal for the study of electrical synapses in neural circuits of genetically-tractable species.

The authors provide evidence of the sensitivity of this method that results from extensive testing under different conditions. Every possible source of confound was explored and addressed. Thus, this method certainly adds to the palette of methods available for studying gap junctional communication and it could be used in combination with other approaches.

A concern of the technique is the modification of intracellular pH resulting from changes in the concentration of protons. As the authors discuss, gap junction channels were shown to be gated by pH. However, the authors document the changes in pH resulting from different activation strengths and conclude that it is possible to operate with sufficient sensitivity and minimal variation in intracellular pH. It is anyway a limitation of the technique that should be used with caution, as the sensitivity for pH might be different for gap junction channels made of different isoforms and the pH buffering capability (although highly conserved) could be different between cells types. While the authors discuss the effect of pH on connexin-based gap junction channels there is not reference to the effect of pH on innexin-based gap junctions, at which this novel approach was also tested and will most likely be used. I suggest the authors should include this point into their Discussion/considerations. Here are a few papers on the effect of pH on invertebrate gap junctions that the authors should include: Giaume, Spira and Korn, 1980; Obaid, Socolar and Rose, 1983; Moreno et al., 1991; Landesman et al., 1999; Anderson and Woodruff, 2001. Anyway, while a limitation (inherent to any method) reminding the experimenter to use this method with caution, it does not weaken the many virtues of the approach.

Overall, this is a new exciting approach that provides a new tool to study the role of electrical synapses in neuronal circuits. The presence, distribution and functional role of electrical synapses has recently received a great deal of attention in *C. elegans* and *Drosophila*, at which neurons are less accessible with electrophysiological approaches. Together with the future technical improvements discussed by the authors its likely immediate use in these genetically tractable invertebrate species will promote the necessary improvements to make possible its use in the vertebrate brain.

References:

Anderson KL, Woodruff RI., A gap junctionally transmitted epithelial cell signal regulates endocytic yolk uptake in Oncopeltus fasciatus. Dev Biol (2001) 239(1):68-78. DOI: 10.1006/dbio.2001.0433

Landesman Y, White TW, Starich TA, Shaw JE, Goodenough DA, Paul DL., Innexin-3 forms connexin-like intercellular channels. J Cell Sci (1999) 112 (Pt 14):2391-6.

Moreno AP, Spray DC, Ramón F., Humoral factors reduce gap junction sensitivity to cytoplasmic pH. I. Organ ablation studies. Am J Physiol (1991) 260(5 Pt 1):C1028-38. DOI: 10.1152/ajpcell.1991.260.5.C1028

*Reviewer #3:*

This study describes a new optical methodology, PARIS, for functionally mapping gap junction synapses between neurons. The approach involves expression of an actuator in one cell (or cell type) and a receiver in another cell (or cell type). The authors tested several possible actuators and receivers and settled on ArchT (a light-gated proton pump) and pHluorinCAAX (a membrane-bound pH sensitive GFP). Photoactivation of ArchT lets protons flow out of the actuator cell, which are then replenished by protons flowing in from another cell, through gap-junctions. If the gap-junction-coupled cell expresses pHluorinCAAX, then the loss of protons changes the pH, which yields a change in fluorescence.

The authors demonstrate that this methodology works in HEK cells, cardiomyocytes, and *Drosophila* neurons ex vivo. They validate it against other established methodologies for detecting gap junctions, specifically with paired patch-clamp recordings or with fluorescence recovery after photobleaching (FRAP). Because PARIS is an optical method, it is easier to perform than paired electrophysiological recordings and has the advantage of subcellular localization of signals. Further, they demonstrate that PARIS is more stable and has higher temporal resolution than FRAP.

Overall, I see this as a useful new methodology in *Drosophila*. I feel that this tool is appropriate for publication in *eLife*, since it promises to facilitate the detection of gap junctions, which could easily enable novel insights into neural circuit functions. I do however, have a couple of substantial concerns that the authors should address.

1) In many instances, researchers may want to apply this technique in vivo. Is the ΔF/F_0_ signal amplitude large enough for this technique to be useful in vivo, where spontaneous activity might mask the small signal? The authors should include either a demonstration of in vivo signal detection, or a frank discussion of limitations of its use in vivo. Potential users of this technique would want to know about these limitations, as this knowledge could potentially save them considerable time when applying it.

2) As with any signal detection question, there is the issue of false positives and false negatives. The authors should provide a practical guide to assessing this issue. For instance, to protect against false negatives (missing a connection when one indeed exists), it seems that a control experiment in the cis configuration is advisable (to ensure that the actuator is having its desired effect). To protect against false positives (finding a connection when one does not actually exist), it seems that washing on CBX as a control is important, especially as a way to confirm that the detected signal is indeed due to a gap junction. The authors should clearly state that these controls are necessary to drawing strong conclusions from this tool (or, if they disagree, clearly explain why). These are important issues, since other tools for detecting synaptic connectivity (e.g. early versions of GRASP in *Drosophila*) were susceptible to false positives (see Shearin et al., 2018, for a discussion). Thus, the authors should address this issue directly.

3) The authors should provide some explanation for why Ca^2+^ based approaches don't work (Figure 1—figure supplement 1B). It is surprising that combining CsChrimson and GCaMP in the same cell (i.e. in the "cis" configuration) does not yield a measurable signal, especially since this general approach has been used successfully in other systems (e.g. Packer et al., 2015).

References:

Packer AM, Russell LE, Dalgleish HW, Häusser M. Simultaneous all-optical manipulation and recording of neural circuit activity with cellular resolution in vivo. Nat Methods. 2015 Feb;12(2):140-6. doi: 10.1038/nmeth.3217.

Shearin HK, Quinn CD, Mackin RD, Macdonald IS, Stowers RS. t-GRASP, a targeted GRASP for assessing neuronal connectivity. J Neurosci Methods. 2018 Aug 1;306:94-102. doi: 10.1016/j.jneumeth.2018.05.014.

---

## [Author Response]

Reviewer #1:[…] If I have one major concern with the utility of PARIS, it is that it seems to be at this point a largely qualitative tool. The authors have demonstrated that functional coupling can be identified with PARIS. What would truly be exceptional is if this could be used to measure not only the presence of coupling, but the relative strength of coupling as well. To do so, the authors would need to measure coupling strength with a traditional approach (e.g. current clamp or voltage clamp), measure with PARIS in the same cells, and correlate this across a number of cell pairs with varying coupling strengths. While I do not think this is required for this study to be ready for publication, it would greatly enhance the utility of this approach.

We agree with the reviewer that PARIS would be valuable if it could detect both the existence of gap junction couplings and the variations of coupling strength. As the reviewer suggested, this could be demonstrated by measuring the coupling strength between the same cell pair using an additional method, such as electrophysiological recording, in coordination with PARIS and correlate these two signals among cell pairs variously connected. Even though we did not perform the exact experiments as the reviewer suggested, as we have shown in Figure 3, PARIS signals were decreased in cell pairs under TPA (PKC pathway agonist) applications and in the cell pairs expressing mutated gap junction proteins compared to the control cell pairs, which may partially indicate PARIS has the potential to distinguish the different coupling strength at least in vitro. While we agree that it would be even better if we could calibrate PARIS by the quantitative patch-clamp method, in order to more accurately compare the coupling strength between different cells/animals.

My only other concern, which is touched on by the authors, is that PARIS will manipulate intracellular pH – and invoke homeostatic mechanisms to maintain cellular pH. If this occurs, there may be physiological questions that alter cellular properties as a result. This may limit the utility of PARIS for longitudinal studies where multiple measurements of coupling are taken over the time course of some manipulation.

We appreciate the reviewer’s concern with respect to pH influences. We have shown that:

1) PARIS signals can be detected when the intracellular pH in actuator cells increased by 0.1 pH unit, which is minimal and brief (Figure 1—figure supplement 4);

2) the ex vivo cell-autonomous signal is stable for as long as 2 hours under up to ten sequential photostimulations with 10 min intervals (Figure 4—figure supplement 1).

Therefore, we speculate that PARIS’s influence on cell physiology is limited and controllable. For the longer measurements, either reduce the power or shorten the time of laser illumination, meanwhile increase the interval between each measurement should be helpful to decrease the pH influence. This concern is also mentioned by reviewer 2 and we have discussed this issue in our revised manuscript (subsection “Limitations of PARIS”, first paragraph).

Reviewer #2:[…] A concern of the technique is the modification of intracellular pH resulting from changes in the concentration of protons. As the authors discuss, gap junction channels were shown to be gated by pH. However, the authors document the changes in pH resulting from different activation strengths and conclude that it is possible to operate with sufficient sensitivity and minimal variation in intracellular pH. It is anyway a limitation of the technique that should be used with caution, as the sensitivity for pH might be different for gap junction channels made of different isoforms and the pH buffering capability (although highly conserved) could be different between cells types. While the authors discuss the effect of pH on connexin-based gap junction channels there is not reference to the effect of pH on innexin-based gap junctions, at which this novel approach was also tested and will most likely be used. I suggest the authors should include this point into their Discussion/considerations. Here are a few papers on the effect of pH on invertebrate gap junctions that the authors should include: Giaume, Spira and Korn, 1980; Obaid, Socolar and Rose, 1983; Moreno et l., 1991; Landesman et al., 1999; Anderson and Woodruff, 2001. Anyway, while a limitation (inherent to any method) reminding the experimenter to use this method with caution, it does not weaken the many virtues of the approach.

We agree with the reviewer that PARIS should be used with caution when it comes to pH modification, since gap junctions consisted of different subunit isoforms might have different sensitivity for pH and the pH buffering capability could vary among cells types. We have added a part in our revised Discussion to remind the researchers to use PARIS with caution (subsection “Limitations of PARIS”, first paragraph).

We appreciate the reviewer for raising a critical point regarding the inadequate discussion about innexin-based gap junctions. We have carefully read the suggested papers and revised the Discussion accordingly (see the aforementioned paragraph).

Reviewer #3:[…] Overall, I see this as a useful new methodology in Drosophila. I feel that this tool is appropriate for publication in eLife, since it promises to facilitate the detection of gap junctions, which could easily enable novel insights into neural circuit functions. I do however, have a couple of substantial concerns that the authors should address.1) In many instances, researchers may want to apply this technique in vivo. Is the ΔF/F0 signal amplitude large enough for this technique to be useful in vivo, where spontaneous activity might mask the small signal? The authors should include either a demonstration of in vivo signal detection, or a frank discussion of limitations of its use in vivo. Potential users of this technique would want to know about these limitations, as this knowledge could potentially save them considerable time when applying it.

We thank the reviewer for pointing out the potential problem of the in vivo application of PARIS. The in vivo signal will be at least related to several aspects, including the expression level of the actuators and receivers, photostimulation efficiency of the actuators, and most of all, the coupling efficiency of the gap junction channels (the opening state as well as the amount of channels displayed on the membrane). In the manuscript, we showed that PARIS could detect eLN-ePN coupling ex vivo with ΔF/F_0_around ~5% and a SNR of nearly 10 to 30. The eLN-ePN coupling efficacy was actually very low (Ju Huang et al., 2010, Figure 3D; Kaiyu Wang et al., 2014, see Discussion) which could probably only otherwise be detected by patch-clamping (the recorded coupling voltage were less than 1mV and have been averaged for 20 trials in the 2010 paper mentioned above). This may to some extend indicate that our PARIS method which generates and detects locally altered pH concentration is already relatively sensitive among the existing methods. While we do suggest that the researchers start with the in cis test in different preparations and contexts and use the cell-autonomous signal to optimize the expression level of actuators/receivers as well as the photostimulation conditions. Besides, based on the in vitro result in Figure 5, the use of the more powerful proton pump Lari, is worth trying. This concern about PARIS’s in vivo sensitivity is also related to the false negative issue in the reviewer’s second point and it has been discussed in our revised manuscript (see below).

2) As with any signal detection question, there is the issue of false positives and false negatives. The authors should provide a practical guide to assessing this issue. For instance, to protect against false negatives (missing a connection when one indeed exists), it seems that a control experiment in the cis configuration is advisable (to ensure that the actuator is having its desired effect). To protect against false positives (finding a connection when one does not actually exist), it seems that washing on CBX as a control is important, especially as a way to confirm that the detected signal is indeed due to a gap junction. The authors should clearly state that these controls are necessary to drawing strong conclusions from this tool (or, if they disagree, clearly explain why). These are important issues, since other tools for detecting synaptic connectivity (e.g. early versions of GRASP in Drosophila) were susceptible to false positives (see Shearin et al., 2018, for a discussion). Thus, the authors should address this issue directly.

We appreciate the reviewer’s critical suggestions and the references about GRASP, which requires a similar trans-expression strategy as PARIS of two components to detect chemical synapses, are very informative and helpful. As the reviewer pointed out, false negative/positive issues are inevitable for any methods, which should be addressed with proper controls that have been demonstrated in the manuscript (Figure 4 and 5, both the in cis test and the CBX blocking controls have been performed in flies). We agree with the reviewer that it is advisable to state more directly about the importance of such controls. We have added this part in our revised Discussion (subsection “Limitations of PARIS”, last paragraph).

3) The authors should provide some explanation for why Ca^2+^ based approaches don't work (Figure 1—figure supplement 1B). It is surprising that combining CsChrimson and GCaMP in the same cell (i.e. in the "cis" configuration) does not yield a measurable signal, especially since this general approach has been used successfully in other systems (e.g. Packer et al., 2015).

We thank the reviewer for raising this point and we have rephrased the reason in the revised manuscript (subsection “Choice of the actuator and receiver”). Similar to the results in the reference mentioned by the reviewer, we have demonstrated that CsChrimson/GCaMP6s could function in cis to generate over 200% ΔF/F_0_of GCaMP6s signals in cultured hippocampal neurons that endogenously express voltage-gated Ca^2+^ channels, which allow further Ca^2+^ influx (Author response image 1). While in non-excitable cells such as HEK293T cells, which do not express voltage gated ion channels, we assume the activation of CsChrimson did allow Ca^2+^ to flow in, however it was still under the detection limit of GCaMP6s, given that CsChrimson is permeable to not only Ca^2+^ but also K^+^, H^+^ and Na^+^, and the permeability of Ca^2+^ is the worst among the four cations (Klapoetke et al., 2014).

**Author response image 1. respfig1:** Cell-autonomous responses in cultured hippocampal neurons co-expressing CsChrimson and GCaMP6s. Scale bar, 30μm.